# Do Users Benefit From Interpretable Vision? A User Study, Baseline, And Dataset

**Leon Sixt**[*1], **Martin Schuessler**[*23], **Oana-Iuliana Popescu**[1], **Philipp Weiß**[3], **Tim Landgraf**[1]

Freie Universität Berlin[1] Weizenbaum Institut Berlin[2] TU Berlin[3]

leon.sixt@fu-berlin.de, martin.schuessler@tu-berlin.de

[*] Equal Contribution

## Abstract

A variety of methods exist to explain image classification models. However, it remains unclear whether they provide any benefit to users over simply comparing various inputs and the model's respective predictions. We conducted a user study (N=240) to test how such a baseline explanation technique performs against concept-based and counterfactual explanations. To this end, we contribute a synthetic dataset generator capable of biasing individual attributes and quantifying their relevance to the model. In a study, we assess if participants can identify the relevant set of attributes compared to the ground-truth. Our results show that the baseline outperformed concept-based explanations. Counterfactual explanations from an invertible neural network performed similarly as the baseline. Still, they allowed users to identify some attributes more accurately. Our results highlight the importance of measuring how well users can reason about biases of a model, rather than solely relying on technical evaluations or proxy tasks. We open-source our study and dataset so it can serve as a blue-print for future studies.

## 1 Introduction

Deep neural networks have been widely adopted in many domains. Yet, for some applications, their use may be limited by how little we understand which features are relevant. Whether engineer or user, insurance company, or regulatory body; all require reliable information about what the model has learned or why the model provides a certain output. Numerous methods have been proposed to explain deep neural networks (Gilpin et al., 2018; Molnar et al., 2020).

Ultimately, to evaluate whether such explanations are helpful, we need user studies (Doshi-Velez & Kim, 2017; Wortman Vaughan & Wallach, 2020). In fact, some studies provided evidence that interpretable ML techniques may be helpful to find biases or spurious correlations (Ribeiro et al., 2016b; Adebayo et al., 2020a). However, a substantial body of work shows that they may not be as helpful as claimed (Kaur et al., 2020; Alqaraawi et al., 2020; Chu et al., 2020; Shen & Huang, 2020). Consequently, it seems that in real-world applications, biases are often found by simply inspecting the model's predictions rather than applying interpretable ML. A recent example is the Twitter image cropping algorithm: it was the users who discovered that it favored white people over people of color (Yee et al., 2021). In this work, we ask: do modern interpretability methods enable users to discover biases better than by simply inspecting input/output pairs?

To investigate this question empirically, we first propose Two4Two: a synthetic dataset depicting two abstract animals. Its data-generating factors can be correlated with the binary target class, thereby creating arbitrarily strong biases. We designed a baseline explanation technique for bias discovery using only the model's output: input images are arranged in a grid grouped by the model's logit predictions. This design allows a user to inspect all attributes that potentially predict the target class.

In an initial user study (N=50), we validated that participants were struggling to find both biases contained in our dataset using this technique. Hence, more elaborate methods can improve over the baseline on this dataset. In the main study (N=240), we compared the baseline against two state-of-the-art explanations: automatically-discovered concepts and counterfactual interpolations generated with an invertible neural network.

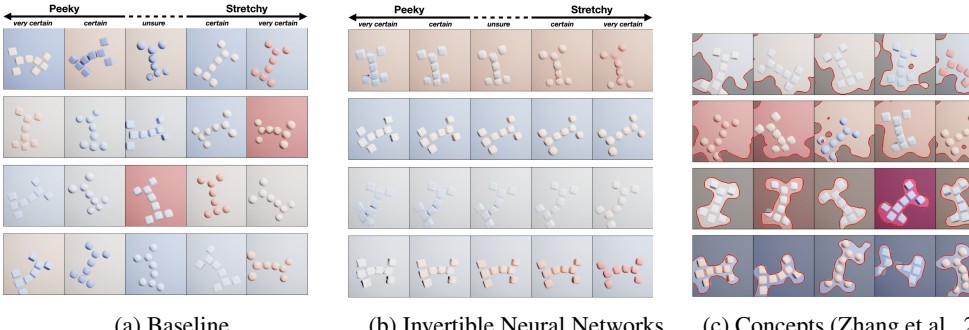

(a) Baseline        (b) Invertible Neural Networks        (c) Concepts (Zhang et al., 2021)

Figure 1: We tested whether users can identify the class-relevant features of images showing two types of animals. We biased attributes like the animal's color to be predictive of the class and investigated whether explanation techniques enabled users to discover these biases. We tested a simple baseline **(a)** which shows random samples grouped by the model's output logit, counterfactual samples generated by an invertible neural network **(b)**, and automatically discovered concepts **(c)**. A participant viewed only one of the above conditions.

We found that none of these explanations outperformed the baseline, even though some features were identified more accurately with counterfactuals. The textual justifications of participants revealed several usability issues in all methods. This highlights the necessity to validate any claims about the benefits of explanation techniques in user studies.

This work represents substantial empirical novelty and significance in the field of interpretable ML:

- The TWO4TWO dataset generator provides control over features and biases. It is designed specifically for human subject studies and to challenge existing interpretability approaches,
- Methods to quantify ground-truth feature importance when the data-generating process is known,
- A study design that provides a unified approach to evaluating interpretable vision methods on the task of bias discovery. It is suitable for lay-users and includes several measures to ensure high-quality crowd-sourced responses,
- A strong and simple baseline explanation technique using only the model output, which we propose as a benchmark for future studies,
- We open-source our dataset, explanation techniques, model, study design, including instructions and videos to support replicating our results as well as adapting our design to other explanation techniques.

## 2   RELATED WORK

**Interpretable ML for Vision**   Different explanation approaches have been proposed: saliency maps (Bach et al., 2015; Ancona et al., 2018; Sundararajan et al., 2017), example-based explanations (Cai et al., 2019), counterfactual examples (Singla et al., 2020), activation-concept approaches (Kim et al., 2018), or models with built-in interpretability (Chen et al., 2019; Brendel & Bethge, 2018). For a detailed review about the field, we refer to (Gilpin et al., 2018; Molnar et al., 2020).

Our work focuses on counterfactual explanations and automatically-discovered concepts. Counterfactual explanations are samples that change the model output, e.g., flip the output class (Wachter et al., 2018). We generated counterfactuals with invertible neural networks (INNs) (Jacobsen et al., 2018; Kingma & Dhariwal, 2018). This approach has recently gained momentum (Hvilshøj et al., 2021; Dombrowski et al., 2021; Mackowiak et al., 2020). Previous works have also used GANs and VAEs for counterfactual generation (Goyal et al., 2019; Mertes et al., 2020; Sauer & Geiger, 2021; Singla et al., 2020; Liu et al., 2019; Baumgartner et al., 2018; Chang et al., 2019). The main advantage of using INNs for counterfactuals is that the generative function is perfectly aligned with the forward function, as an analytic inverse exists.

Concepts represent abstract properties, which can be used to explain a model. For example, the classification of an image as "zebra" could be explained by a pronounced similarity to the "stripe" concept. This similarity is determined by the dot product of the network's internal activations with a concept vector. TCAV (Kim et al., 2018) required manually defined concepts. Recent works proposed to discover concepts automatically (Ghorbani et al., 2019; Zhang et al., 2021).

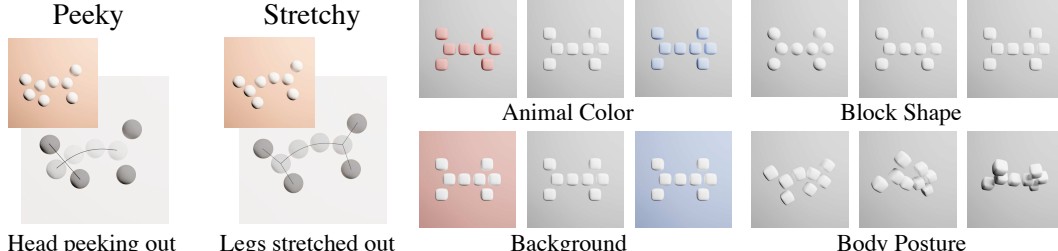

Figure 2: The left panel depicts the main difference between *Peeky* and *Stretchy*: the legs' position. While *Peeky* shows one pair of legs moved inwards, *Stretchy's* legs are moved outwards. Two4Two offers different attributes: animal color, background color, the shape of the blocks and the animal's body posture. All of which can be controlled and biased separately.

**User Studies for Interpretability**  Previous works with the task of bias discovery have mainly evaluated saliency maps and used datasets with a single, simple bias, e.g. background Adebayo et al. (2020a); Ribeiro et al. (2016a) or image watermarks Kim et al. (2018). User studies for concept-based methods tested only the accessibility of the explanations by asking users to assign images to a concept (Zhang et al., 2021; Ghorbani et al., 2019). Counterfactual explanations have been evaluated by Mertes et al. (2020) on a forward-prediction task. We thus believe that we are the first to extensively test counterfactual-based and concept-based explanations on bias discovery using a challenging dataset. Recently, a study on exemplary-based explanations focused on understanding internal activations of a neural network (Borowski et al., 2020). It showed that for this task, examples could be more beneficial than complex feature visualizations (Olah et al., 2017). Similarly, there is evidence that participants often rely on model predictions rather than on explanations (Alqaraawi et al., 2020; Adebayo et al., 2020a).

**Synthetic Datasets for Interpretable Vision**  Datasets with known ground-truth biases have been proposed before. BAM is an artificial dataset (Yang & Kim, 2019) where spurious background correlations are introduced by pasting segmented objects on different textures, e.g. dogs on bamboo forests. However, the resulting images are unsuitable for user studies as they look artificial and make it easy for participants to suspect that the background is important. Additionally, it would be difficult to introduce more than one bias. A limitation that also the synthetic dataset in (Chen et al., 2018) shares. The synthetic dataset created by Arras et al. (2021) created a dataset to technically evaluate saliency methods on a visual question answering task technically. Two4Two is the first dataset designed explicitly for human subject evaluations. To the best of our knowledge, we provide the first unified approach to evaluate interpretable vision on a bias-discovery task.

# 3    TWO4TWO: DATASETS WITH KNOWN FEATURE IMPORTANCE

**Dataset Description**  Datasets generated with Two4Two consist of two abstract *animal* classes, called *Peeky* and *Stretchy*. Both consist of eight blocks: four for the spine and four for the legs. For both animals, one pair of legs is always at an extended position. The other pair moves parallel to the spine inward and outward. The attribute *legs' position*, a scalar in [0,1], controls the position. At a value of 0.5, the pair of legs are at the same vertical position as the last block of the spine. Peekies have a leg position $\leq 0.52$ which means legs are moved mostly inwards to the body center. In the same fashion, Stretchies are extended outwards, legs' position $\geq 0.48$. We added some ambiguity to ensure a model has an incentive to use possible biases. Therefore, Peekies and Stretchies are equally likely for a legs' position between 0.48 and 0.52. It is also difficult for humans to tell if the legs are outward or inwards in this range. Besides the legs' position, the dataset has the following parameters: *body posture* (bending and three rotation angles), *position*, *animal color* (from red to blue), *blocks' shape* (from cubes to spheres), and *background color* (from red to blue). Each can be changed arbitrarily and continuously (See Appendix Table 5).

When designing the dataset, we wanted to ensure that (1) participants can become experts within a few minutes of training, (2) it allows for the creation of multiple biases that are difficult to find, and (3) that it provides a challenge for existing interpretability methods. Goal (1) is met as participants can be instructed using only a few examples (see the tutorial video in Appendix C). The high number of controllable attributes achieve Goal (2). We biased the attributes such that they do not stand out, which we validated in the first user study. Goal (3) is met by spatially overlapping attributes and long-

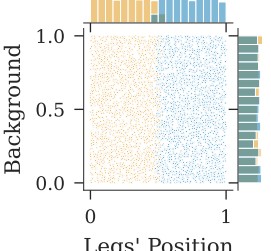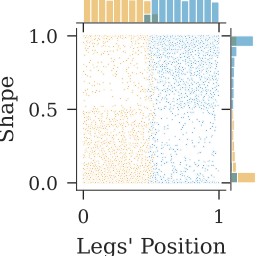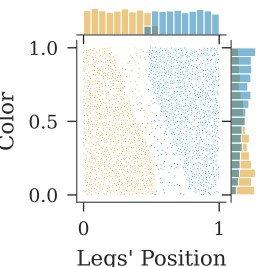

Figure 3: The joint distributions of legs' position and the attributes background (left), shape (middle), and color (right). Datapoints are yellow for Peekies and blue for Stretchies. The background is not biased. The shape is biased for legs' position lower than (0.45) or greater (0.55), but is uniform in the center. The color contains additional predictive information about the target class, as it allow to discriminate between Peeky and Stretchy where the legs' position overlaps. However, for more extreme arms' positions the color is uniform and not biased.

range image dependencies. Spatially overlapping attributes, like color and shape, directly challenge saliency map explanations. Long-range image dependencies, like the legs' positions relative to the spine, can not be explained when analyzing patches separately as done in (Chen et al., 2019; Brendel & Bethge, 2018). Both properties are common in real-world datasets: For example, race and gender in facial datasets are encoded by spatially overlapping features. Long-range image dependencies are particularly relevant for pose estimation and visual reasoning (Johnson et al., 2017).

**Introducing Biases**  For our studies' dataset, we sampled the block's shape in a non-predictive biased fashion. This means that for legs' positions that clearly showed a Peeky [0, 0.45] most blocks were rather cubic, while for legs' positions that clearly showed a Stretchy [0.55, 1] most blocks were rather round. However, for the legs' positions between [0.45, 0.55] the blocks shape was uniformly distributed. In particular, in the even narrower interval [0.48, 0.52] where a classifier can only be as good as random guessing, the block's shape does not provide any additional information about the target class. In Figure 3, we show the joint distribution of the block's shape and legs' position.

We sampled the animals' color to be predictive for the target class. At the small interval where the legs overlap [0.48; 0.52], we distributed the animal color to provide additional class information. Stretchies were more likely to be red, and Peekies were more likely to be blue. Outside of this centered interval, the color gradually became uniformly distributed (see Figure 3). Hence, color was more equally distributed than the shape, making the color bias harder to detect visually. The remaining attributes, background color and body posture, were sampled independently of the class, and we expected our model to ignore them.

**Measuring Ground-Truth Feature Importance**  Even if a dataset contains biases, it is unclear how relevant they will be to a neural network after training. Feature importance also depends on the network architecture, the optimization process, and even the weight initialization. As Two4Two allows us to change any parameter in isolation, we can directly compare the model prediction between two images that differ in only one parameter. For these two images, we measured both the median absolute logit change and also for how many samples the predicted class was flipped. Both measures quantify how influential each parameter is (see Table 1). As expected, the legs' position had a strong influence on the prediction. The model relied more on animal color than on the blocks' shape, which is expected as the color contains additional information about the class. Surprisingly, the prediction flip for unrelated attributes such as background was only slightly lower than for blocks' shape.

To analyze this further, we calculated a linear fit for each parameter change to the logit change. We reported the coefficient of determination $R^2$, which indicates how much of the variance in the prediction can be explained linearly by the analyzed property. While the unrelated properties sometimes flip a prediction, the direction of that flip is random ($R^2 \approx 0$). In contrast, the biased parameters influence predictions in a directed fashion, with animal color ($R^2$=0.751) being clearly more directed than blocks' shape ($R^2$=0.307).

## 4    MODEL AND EVALUATED METHODS

As discussed in section 3, Two4Two was designed to challenge existing interpretability methods, e.g., saliency map explanations and patch-based models. We selected two methods that might provide

Table 1: Importance of the data generating factors to the model's prediction. *Prediction Flip* quantifies how often the model's prediction changes the sign when changing the attribute. The *Mean Logit Change* reports the median of the absolute change in logit values. The $R^2$ score is calculated on an ordinary least squares from the changes of each factor to the changes in the model's logit. For more attributes, see Appendix Table 3.

| Factor | Prediction Flip [%] | Median Logit Change | $R^2$ |
|---|---|---|---|
| Legs' Position | 41.680 | 2.493 | 0.933 |
| Color | 7.080 | 0.886 | 0.751 |
| Shape | 3.920 | 0.577 | 0.307 |
| Background | 2.640 | 0.523 | 0.006 |
| Rotation Yaw | 3.480 | 0.669 | 0.001 |
| Bending | 3.640 | 0.605 | 0.000 |

the user with the necessary information: counterfactuals generated with an invertible neural network (INN) and concept-based explanations (Zhang et al., 2021).

**INN Counterfactuals**   We trained an INN using both a supervised and an unsupervised objective (Dinh et al., 2016; 2015). To predict the target class, the model first applies the forward function $\varphi$ to map a data point $x$ to a feature vector $z = \varphi(x)$. Then, a linear classifier takes those features $z$ and predicts the logit score $f(x) = w^T z + b$. Any input can be reconstructed from the feature vector by applying the inverse function $x = \varphi^{-1}(z)$. The model has a test accuracy of 96.7%. Further details can be found in Appendix A.2. The baseline and concept techniques are also applied to this model.

To create a counterfactual example $\tilde{x}$ for a data point $x$, we can exploit the linearity of the classifier. Moving along the weight vector $w$, i.e., adding $w$ to the features $z$, changes the model's prediction. By controlling the step size with a scalar $\alpha$, we can directly quantify the change in the logit value $\Delta y = \alpha w^T w$. The modified feature vector $z + \alpha w$ can be inverted back to the input domain, resulting in a counterfactual $\tilde{x} = \varphi^{-1}(z + \alpha w)$ which visualizes the changes introduced by a step $\alpha w$ in $z$-space. The INN's explanations are visualized in a grid where each row shows a single counterfactual interpolation (see Figure 1b).

**Automatically-discovered Concepts**   We adapted the NMF approach of Zhang et al. (2021) to our specific network architecture. Because the network's internal representations also contain negative values, we used matrix factorization instead of NMF. We generated the concepts using layer 342 (from a total of 641 layers). The layer has a feature map resolution of 8x8. This choice represents a trade-off between enough spatial resolution and high-level information. We ran the matrix factorization with 10 components and selected the five components that correlated most with the logit score ($r$ is in the range [0.21, 0.34]).

Our presentation of concept-based explanations was very similar to (Zhang et al., 2021): we visualized concepts with five exemplary images per row and highlighted regions corresponding to a concept. Since our classifier is binary, a negative contribution for Stretchy actually means a positive contribution for Peeky. Hence, we could have characterized a concept as *more Peeky* and *more Stretchy*, to make the design similar to the other two explanation techniques. However, as the concepts did not strongly correlate with the model's output, presenting them as class-related could confuse participants: a *more Peeky* column would have contained some images showing Stretchies and vice versa. Thus, we presented them separately in two consecutive rows (See Figure 1c). Presenting concepts in this fashion gives them a fair chance in the study because participants rated the relevance of each attribute for the model rather than for each class separately.

## 5   HUMAN SUBJECT STUDY

We share the view of Doshi-Velez & Kim (2017) and Wortman Vaughan & Wallach (2020) that user-testing of explanation techniques is a crucial but challenging endeavor. As our second main contribution, we propose and conduct a user study based on the Two4Two dataset which can act as a blue-print for future investigations. Our design has been iterated in over ten pilot studies and proposes solutions to common problems that arise when evaluating explanation techniques on crowd-sourcing platforms with lay participants.

## 5.1 Design considerations

**Data without Prior Domain Knowlege** We specifically designed the Two4Two dataset to avoid overburdening participants, as might be the case with other types of data. Within a few minutes, participants can easily become domain experts. Since the data is unknown to them prior to the study, we avoid introducing any prior domain knowledge as a confounding factor, which can be an issue (Alqaraawi et al., 2020).

**Manageable but not Oversimplified Tasks** We propose the task of *bias-discovery*: participants had to rate features as either *relevant* or *irrelevant* to a model. The task directly reflects users' perception of feature importance. Furthermore, bias-discovery has the advantage of being suitable for lay participants. At the same time, it is also grounded in the model's behavior. This is an advantage over tasks used in several previous studies, which only evaluated whether explanations were *accessible* to users, e.g. by identifying the target property *smiling* using image interpolations (Singla et al., 2020) or assigning images to a concept class (Zhang et al., 2021; Ghorbani et al., 2019). However, these tasks are an oversimplification and cannot measure any insights the users gained about the model. In contrast, Alqaraawi et al. (2020) employed the task of *forward prediction* of a neural network. This requires substantial model understanding and is very challenging, as reflected by the participants' low accuracy. Assessing trust in a *human-in-the-loop task*, despite its realistic setting, has the disadvantage that trust is influenced by many factors which are difficult to control for (Lee & See, 2004; Springer et al., 2017). Another approach is to asks participants to assess whether a model is fit for deployment (Ribeiro et al., 2016b; Adebayo et al., 2020b). However, in our own pilots studies, users deemed a model fit for deployment even if they knew it was biased.

**Baseline Explanation Technique** To quantify whether an explanation is beneficial for users, it must be compared to an alternative explanation. In this work, we argue that a very simple and reasonable alternative for users is to inspect the model's logits assigned to a set of input images. We designed such a baseline explanation as shown in Figure 1a. After several design iterations, we settled for a visually dense image grid with 5 columns sorted by the logit score, each column covering 20% of the logit values. The columns were labeled very certain for Peeky/Stretchy, certain for Peeky/Stretchy, and as unsure. Pilot studies showed that participants' attention is limited. We thus decided to display a total of 50 images, i.e. an image grid of 10 rows. The number of images was held constant between explanation techniques to ensure the same amount of visual information and a fair comparison. In this work, we focused on binary classifications. For a multi-class setting, one could adapt the baseline by contrasting one class verses another class.

**High Response Quality** We took extensive measures to ensure participants understood their task and the explanation techniques. Participants were required to watch three professionally-spoken tutorial videos, each under four minutes long. The videos explained, on a high level, the Two4Two dataset, machine learning and how to use an assigned explanation technique to discover relevant features. To avoid influencing participants, we prototyped idealized explanations using images from Two4Two. The explanations showed different biases than those in the study. Each video was followed by a written summary and set of multiple choice comprehension questions After failing such a test once, participants could study the video and summary again. When failing a test for a second time, participants were excluded from the study. We also excluded participants if their written answers reflected a serious misunderstanding of the task, indicated by very short answers copied for all attributes or reasoning that is very different from the tutorial. We recruited participants from Prolific who are fluent in English, hold an academic degree and have an approval rate of $\geq 90\%$. To ensure they are also motivated, we compensated them with an average hourly pay of £11.45 which included a bonus of £0.40 per correct answer.

## 5.2 Experimental Design

We conducted two online user studies. Before starting the data collection, we formulated our hypotheses, chose appropriate statistical tests, and pre-registered our studies (see Appendix D). This way, we follow the gold-standard of defining the statistical analysis before the data collection, thus ensuring that our statistical results are reliable (Cockburn et al., 2018). The first study (N=50) analyzed whether the task was challenging enough that other methods could potentially improve over the baseline. We tested if at least one bias in our model (either the animal's color or the blocks' shape) was difficult to find using the baseline technique. Consequently, we used a within-subject design.

Table 2: The mean accuracy for each attribute by condition. $N_{\text{collected}}$ provide the number of participants collected and $N_{\text{filtered}}$ the number of remaining participants after the filtering. Stars mark statistical significance.

| Condition | $N_{\text{collected}}$ | $N_{\text{filtered}}$ | Overall | Legs | Color | Backgr. | Shape | Posture |
|---|---|---|---|---|---|---|---|---|
| Study 1 (Baseline) | 50 | 43 | 73.4 | 86.0 | 48.8 | 86.0 | 74.4 | 72.1 |
| Study 2 | 240 | 192 | 67.0 | 78.2 | 58.9 | 66.8 | 73.1 | 59.1 |
| INN | 80 | 62 | **84.5** | ***100.0** | *82.3 | *79.0 | 90.3 | **71.0** |
| Baseline | 80 | 71 | 80.8 | 85.9 | 59.2 | **95.8** | **93.0** | 70.4 |
| Concepts | 80 | 59 | 32.2 | 45.8 | 32.2 | 18.6 | 32.2 | 32.2 |

In the second study (N=240), we evaluated the two explanation techniques described in Section 4 against the baseline using a between-subjects design. Participants were randomly, but equally assigned to one of the explanation techniques. We specified two directed hypotheses. We expected participants in the INN condition to perform better than those in baseline, because the baseline does not clearly highlight relevant features, whereas interpolations highlight features in isolation. We expected participants viewing concepts to perform worse than those in the baseline, due to their inability to highlight spatially overlapping features.

For both studies, participants completed a tutorial phase first. Using their assigned explanations, they then assessed the relevance of five attributes: legs' position relative to the spine, animal color, background, rotation or bending, and blocks' shape. The questions were formulated as: *"How relevant is <attribute> for the system?"*, and participants had to choose between *irrelevant* or *relevant*. The percentage of correct answers (accuracy) served as our primary metric. Participants also had to write a short, fully-sentenced justification for their answers. For links to the study, see Appendix C.

## 5.3 RESULTS

**Data Exclusions** As stated in the preregistration, we automatically excluded all participants who withdrew their consent, failed one of the comprehension questions twice, skipped a video, or exceeded Prolific's time limit for completion. If a participant was excluded, a new participant's place was made available until the pre-registered number of completed responses was reached. We excluded 63 study respondents for the first study, and 145 for the second study in this fashion. We ensured that all participants were naive about the dataset. Once they participated in a study, they were blacklisted for future studies.

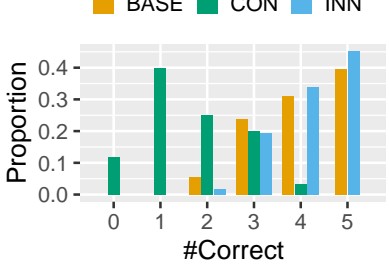

Figure 4: The proportion of correct answers for baseline (BASE), concepts (CON), and INN.

For completed studies, two annotators independently marked the participants' written answers and excluded those with copy and paste answers or indications of grave misunderstandings of the instructions. Participants were labeled as: *include*, *unsure*, or *exclude*. Both anotators had an agreement of $\kappa = 0.545$ for the first study and $\kappa = 0.643$ for the second (measured *include* vs. *unsure* and *exclude*). Disagreements were solved by discussion. In total, we excluded 7 participants from the first study (14%) and 48 participants from the second study (20%).

**First Study** For the accepted 43 participants, we used two-sided exact McNemar tests on their answers about the relevance of the *legs position* compared with *animal color* (first test) and *background* (second test). Participants found the color bias less often than the legs' positions ($P < 0.0001$). The success rate for the color attribute was 49% vs. 86% for legs. The shape bias was not significantly harder to find than the legs' positions and was identified correctly with 74% accuracy ($P = 0.3036$). Hence, we confirmed our hypothesis and concluded that other methods still have room for improvement over the baseline.

**Second Study** In the second study, we evaluated 192 valid participant responses (62 INN, 71 BASE, 59 CON). We expected data to be different from the normal distribution, and a Shapiro-Wilk test for all conditions confirmed this ($P < 0.001$). We depict the number of correct answers per condition in Figure 4. A Kruskal-Wallis test showed a significant differences in accuracy scores between conditions ($P < 0.001$). For focused comparisons, we used two Wilcoxon-rank-sum

tests with Bonferroni correction to correct for multiple comparisons. The accuracy scores differed significantly between the baseline and concept conditions ($P < 0.001$, $r=0.778$). The performance of participants using concepts was rather poor, with only 31.7% accuracy, considering that random answers would yield a score of 50%. For concepts, not a single attribute surpassed the 50% barrier. We found no significant difference when comparing the baseline and counterfactuals ($P=0.441$, $r=0.091$). Their mean accuracies are close, with 80.8% for baseline and 84.5% for counterfactuals. INN counterfactuals helped users to discover the main attribute, legs' position, ($P < 0.001$) and color bias ($P=0.033$) more reliably.[1] However, counterfactuals performed significantly worse for the background attribute ($P=0.033$), while for blocks' shape and position we found no significant difference (for both, $P=1$).

**Qualitative Results** To understand how participants integrated the explanation techniques into their reasoning, we analyzed the textual answers of each feature qualitatively. Two annotators first applied open coding to the answers. They performed another pass of closed coding after agreeing on a subset of the relevant codes, on which the following analysis is based. Overall, the participants perceived the task as challenging, as they expressed being unsure about their answers (N=71).

We designed our image grid to show both possible classes and provide information about the model's certainty. We found that many participants integrated this additional source of information into their reasoning. This was especially prevalent in the baseline condition (N=51). Participants particularly focused on the columns 'very certain Peeky' and 'very certain Stretchy', as well as on the column 'unsure'. While this may have helped confirm or reject their own hypotheses, it sometimes led to confusion; for example, when an image that exhibited a pronounced leg position, and therefore could easily be identified as Peeky or Stretchy, was classified by the model as 'unsure' (N=14).

Across conditions, we also observed that participants expect that all images needed to support a hypothesis. *"The animals are in different colors, there are blue stretchy and also blue peeky animals, If the color was relevant peeky/stretchy would be in one color etc"* (P73, BASE). Across conditions, most participants that applied such deterministic reasoning failed to find the color bias. In contrast, other participants applied more probabilistic reasoning, which helped them deal with such contradictions: *"Peeky is more likely to be blue in colour, whereas Stretchy is more likely to be pink. This is not always true (e.g. the shapes can be white in colour at either ends of the spectrum) but it might be somewhat relevant to help the system decide"* (P197, INN).

Another observed strategy of participants was to reference how often they saw evidence for the relevance of a feature (N=35), which was very prevalent in the concepts condition (N=20). Especially concepts were rather difficult for participants to interpret. A common issue was that they expected a relevant feature to be highlighted completely and consistently (N=38). Several instances show that participants struggled to interpret how a highlighted region can explain the relevance of a feature, *"If this [the legs position] were relevant I would have expected the system to highlight only the portion of the image that contains the legs and spine. (e.g. only the legs and one block of the spine at each end). Instead, every image had minimally the entire animal highlighted"* (P82, CON). Furthermore, spatially overlaping features were another cause of confusion: *"there are rows in which the animal is highlighted but not the background so it could be because of color, shape or rotation"* (P157, CON)

Participants erred more often for the background in the INN condition than for the baseline. We conducted an analysis to investigate this issue. We found that 29 participants stated that they perceived no changes in the background of the counterfactuals and hence considered this feature irrelevant. Another 21 participants noted that they saw such a change, which let 12 of them to believe its a relevant feature. *"The background color changes in every case, it's also a little subtle but it does"* (P205). Another 9 participants decided that the changes were too subtle to be relevant. *"The background colour does not change an awful lot along each row, maybe in a couple of rows it changes slightly but I do not feel the change is significant enough that this is a relevant factor in the machine decision"* (P184).

**Do Counterfactuals Highlight Irrelevant Features?** Indeed, subtle perceptual changes in background color are present (Figure 1b). To quantify these changes, we decided to use an objective observer: a convolutional neural network. We trained a MobileNetV2 (Sandler et al., 2018) to predict

---

[1]The statistical analysis of the attributes for INN vs. baseline was not pre-registered. The reported p-values for the attributes were corrected for eight tests (including the pre-registered tests) using the Holm–Bonferroni method.

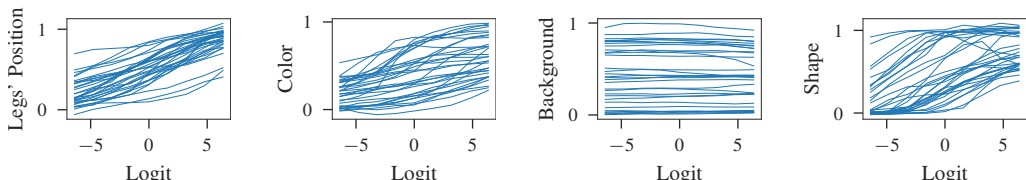

Figure 5: Attribute changes along counterfactual interpolations as measured by an observer convnet. Each line corresponds to a single sample whose logit score is modified through linear interpolations in the classifier space.

the parameter values of individual attributes of an image (e.g., background color, object color, etc.) using a completely unbiased version of TWO4TWO. After training, the model could predict the parameter values almost exactly (MSE < 0.0022, see Table 7). We then used this model to evaluate the parameter values of counterfactual INN interpolations, each spanning 99% of the logit distribution. We visualize the predictions of MobileNetV2 in Figure 5. All predictive properties (legs' position, body color, blocks' shape) are changed by the counterfactuals consistently. For the background, the changes are subtle but present. We also quantified the change in parameters using the difference between the maximum and minimum predicted value per individual interpolation (See Table 6). This supports the finding that relevant attributes change the most – legs' position: 0.662; shapes: 0.624; color: 0.440. The background changes less with 0.045, which seems enough to give some participants a false impression about its relevance.

# 6 CONCLUSION

**Contributions** We contribute a dataset with full control over the biases it contains and methods to verify the feature importances of a given model. The dataset was specifically designed for user studies and we contribute a carefully crafted study design as a template for future empirical evaluations of interpretable vision. Our design includes a simple, yet powerful baseline technique that relies on the model's outputs only. While interpretability methods had room to improve over the baseline, we showed that two state-of-the-art methods did not perform significantly better. Our results emphasize that any explanation technique needs to be evaluated in extensive user studies.

**Limitations** Due to budget constraints, we limited the number of factors in our experimental design (external vs. internal validity trade-off). Our study introduced a predictive bias for the animal's color and a non-predictive bias for the blocks' shape. It remains unclear how our results may have changed for a different dataset configuration: certain biases could exhibit different visual saliency. It remains also left for future work to determine which visual interface design is optimal for a given method. Furthermore, our study design restricted participants to make binary choices and provide textual justifications – limiting our understanding of the participants issues.

**Take-Aways** We were surprised by the performance of the two tested techniques. Users had problems interpreting the automatically-discovered concepts and could not identify the relevant attributes. As we expected, explaining spatially overlapping features by highlighting important regions limited the concepts' expressiveness. On the other hand, INN counterfactuals also did not perform significantly better than the baseline. Still, counterfactuals were more helpful to discover the strongest bias in the model. However, some participants rated the relevance of the background incorrectly, as slight changes in the interpolations were still salient enough. It is therefore important for future work to develop counterfactuals that alter only relevant attributes.

We presented a user study on a synthetic dataset. We believe that the results also have implications for natural image data. When we created Two4Two, our objective was to translate challenges faced on "real" computer vision data (like spatially overlapping features) into an abstract domain. Although some properties of photorealistic datasets are lost in this abstraction, a method performing poorly on TWO4TWO would likely not perform well on a natural dataset with spatially overlapping features.

**Outlook** The user study was a reality check for two interpretability methods. Such studies can guide technical innovations by identifying areas where users still struggle with current explanation methods. They are laborious and expensive, but at the same time, they are crucial for future interpretability research. Future work could focus on creating a more realistic, controllable dataset, e.g. using augmented reality (Alhaija et al., 2018). By open-sourcing our videos, study, and code we encourage the community to take on the challenge to beat the simple baseline.

## 7 ACKNOWLEDGMENTS

We thank Jonas Köhler and Benjamin Wild for their feedback on the manuscript. We also thank our reviewers for their time. LS was supported by the Elsa-von-Neumann Scholarship of the state of Berlin. MS and PW were funded by the German Federal Ministry of Education and Research (BMBF) - NR 16DII113. In addition, the work was partially funded by the the German Federal Ministry of Education and Research (BMBF), under grant number 16DHB4018. We thank the Center for Information Services and High Performance Computing (ZIH) at Dresden University of Technology and the HPC Service of ZEDAT, Freie Universität Berlin, for generous allocations of computation time (Bennett et al., 2020).

## 8 ETHICS STATEMENT

In our human subject evaluation, it was important for us to pay crowd workers £11.45 per hour which is above UK minimum wage of £8.91. The crowd workers consented to the use of their answers in the study. No personal-data was collected. Our organization does not require an approval of online user studies through an ethics review board.

## 9 REPRODUCIBILITY STATEMENT

Our work brings some additional challenges to reproducibility beyond compared to other machine learning research. The results of the human subject study depend on the dataset, model and even the presentations in the video tutorials. We decided to tackle this challenge by first open-sourcing our dataset, model, videos, code, and even the study itself. For now, we share the links to the study and videos in the Appendix. We will share an machine-readable export of the study, the source-code, and the model with our reviewers via OpenReview once the discussion period start and will make them public at a later point in time.

In total, we spent around 5000 GBP including prestudies, software licenses, and our two studies in the paper. We estimate the costs of reproducing the studies in our work at around 2500 GBP excluding software licenses. For a study comparing only one condition against the baseline, we would estimate the costs to be around 1400 GBP.

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

Table 3: Importance of the data generating factors to the model's prediction. For the $R^2$ score, we fitted an ordinary least squares from the factors' deltas to the deltas of the model's logits and then report the coefficient of determination ($R^2$). The *Mean Logit Change* reports the median of the absolute change in logit values. The *Prediction Flip* column quantifies how often the model's prediction changed the sign when changing the attribute.

| Factor | Prediciton Flip [%] | Median Logit Change | $R^2$ |
|---|---|---|---|
| Legs' Position | 41.680 | 2.493 | 0.933 |
| Color | 7.080 | 0.886 | 0.751 |
| Shape | 3.920 | 0.577 | 0.307 |
| Position Y | 2.960 | 0.597 | 0.007 |
| Background | 2.640 | 0.523 | 0.006 |
| Rotation Yaw | 3.480 | 0.669 | 0.001 |
| Rotation Roll | 2.260 | 0.413 | 0.001 |
| Bending | 3.640 | 0.605 | 0.000 |
| Rotation Pitch | 3.500 | 0.627 | 0.000 |
| Position X | 3.380 | 0.581 | 0.000 |

Table 4: *Norm* quantifies the length of feature map changes ($\Delta \boldsymbol{h} = f(x) - f(\hat{x})$) after resampling the different data generative factors. *Angle w. Clas.* quantifies the mean angle (in degrees) between $\Delta \boldsymbol{h}$ and the classifier weight $\boldsymbol{w}$.

| Factor | Norm | Norm Std. | Angle w. Clas. | Angle Std. |
|---|---|---|---|---|
| Legs' Position | 70.8 | 2.6 | 79.9 | 1.9 |
| Color | 53.8 | 2.4 | 85.3 | 1.3 |
| Shape | 66.9 | 2.1 | 88.5 | 1.5 |
| Position Y | 68.1 | 2.8 | 89.7 | 1.7 |
| Background | 60.3 | 2.6 | 89.7 | 1.5 |
| Bending | 68.9 | 2.6 | 89.8 | 1.7 |
| Rotation Yaw | 68.5 | 2.7 | 89.8 | 1.7 |
| Rotation Roll | 55.2 | 2.9 | 89.9 | 1.2 |
| Position X | 68.4 | 3.0 | 89.9 | 1.6 |
| Rotation Pitch | 69.5 | 2.5 | 90.0 | 1.5 |

## A  APPENDIX: TECHNICAL DETAILS

### A.1  TWO4TWO DATASET DETAILS

| Factor | Range | Distribution | Biased | Additional Class Information |
|---|---|---|---|---|
| Legs' Position | $[0, 1]$ | Uniform with overlap | Yes | - |
| Color | $[0, 1]$ | See Figure 3 | Yes | Yes |
| Shape | $[0, 1]$ | See Figure 3 | Yes | No |
| Position Y | $[-0.8, 0]$ | Uniform | No | No |
| Position X | $[-0.8, 0]$ | Uniform | No | No |
| Background | $[0.05, 0.95]$ | Uniform | No | No |
| Rotation Yaw | $[0, 2\pi]$ | Uniform | No | No |
| Rotation Roll | $[-\pi/4, \pi/4]$ | Truncated Normal$(0, 0.03\pi/4)$ | No | No |
| Rotation Pitch | $[-\pi/6, \pi/6]$ | Truncated Normal$(0, \pi / 8)$ | No | No |
| Bending | $[-\pi/10, \pi/10]$ | Truncated Normal$(0, \pi / 20)$ | No | No |

Table 5: Distribution of each attribute in the study's dataset. *Biased* denotes whether an attribute is unequally distributed for the two classes. *Additional Class Information* show if an attribute contains any additional information about the target class not already given by the legs' position.

Table 6: Two4Two: Effect of interpolating along the weight vector.

| Attribute | Mean Maximal Change | Std. |
|---|---|---|
| Legs' Position | 0.662 | 0.140 |
| Color | 0.440 | 0.190 |
| Shapes | 0.624 | 0.208 |
| Bending | 0.059 | 0.042 |
| Background | 0.045 | 0.044 |
| Rotation Pitch | 0.186 | 0.126 |
| Rotation Yaw | 0.102 | 0.182 |
| Rotation Roll | 0.003 | 0.001 |
| Position X | 0.105 | 0.078 |
| Position Y | 0.103 | 0.078 |

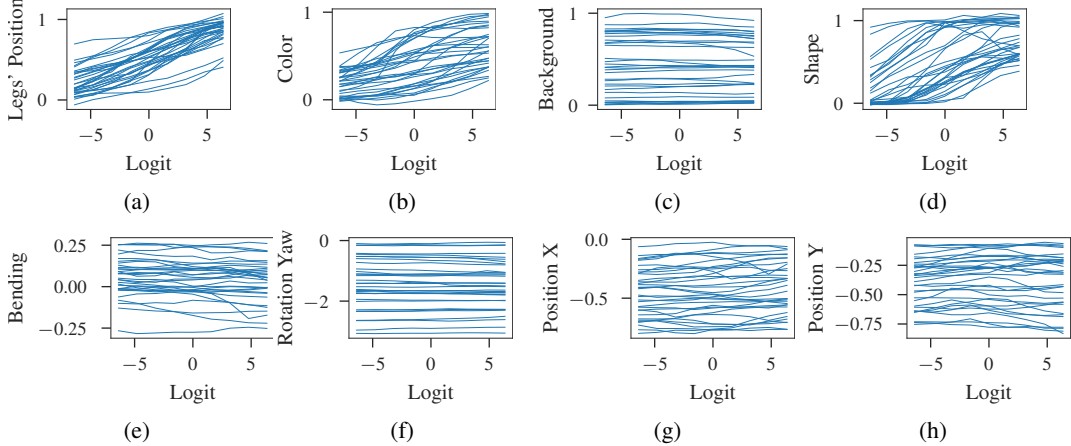

Figure 6: All attribute values as predicted by an observer convnet for sequences of counterfactual interpolations. Each line corresponds to a single sample whose logit score is modified through linear interpolations in classifier space.

```python
import dataclasses
import numpy as np
import matplotlib.pyplot as plt

from two4two.blender import render
from two4two.bias import Sampler, Continouos
from two4two.scene_parameters import SceneParameters

@dataclasses.dataclass
class RotationBiasSampler(Sampler):
    """A rotation-biased sampler.

    The rotation is sampled conditionally depending on the object type.
    Positive rotations for peaky and negative rotations for stretchy.
    """

    obj_rotation_yaw: Continouos = dataclasses.field(
        default_factory=lambda: {
            'peaky': np.random.uniform(-np.pi / 4, 0),
            'stretchy': np.random.uniform(0, np.pi / 4),
        })

# sample a 4 images
sampler = RotationBiasSampler()
params = [sampler.sample() for _ in range(4)]
for img, mask, param in render(params):
    plt.imshow(img)
    plt.title(f"{param.obj_name}: {param.obj_rotation_yaw}")
    plt.show()
```

Listing 1: Source code example to create a biased sampler. High positive rotations are predictive of Stretchy and low negative rotations of Peaky.

## A.2 ARCHITECTURE OF THE INVERTIBLE NEURAL NETWORK

Our model is based on the Glow architecture (Kingma & Dhariwal, 2018) and contains 7 blocks. A block is a collection of 32 flow steps, followed by a down-sampling layer, and ends with a fade-out layer. A single flow step consists of *actnorm*, *invertible* $1 \times 1$ *convolution* and *affine coupling* layer. The down-sampling keeps all dimensions, e.g. a shape of $(h, w, c)$ becomes $(h/2, w/2, 4c)$. The fade-out layer maps removes half of the channels. The out-faded channels are than mapped to a standard normal distribution to compute the unsupervised loss. For generating counterfactuals, the out-faded values are not thrown away but rather stored to be used when computing the inverse.

The model is trained using a supervised loss and an unsupervised objective. In total our model had 687 layers and 261 million parameters. The classifier used the output of layer 641. The remaining layers 642-687 were optimized using the standard unsupervised flow objective. For the first 641 layers, we also trained on the classifier's supervised loss.

Let $\varphi$ denote the first 641 layers and $\mu : \mathbb{R}^n \mapsto \mathbb{R}^n$ the last. We train $\varphi$ both on a supervised loss from the classifier $f(\boldsymbol{x})$ and an unsupervised loss from matching the prior distribution $\mathcal{N}(0, I)$ and the log determinante of the Jacobian. $\mu$ is only trained on the unsupervised loss:

$$\arg \min_{\theta_\varphi, \theta_\mu, \theta_f} L_{\text{un}}(\mu \circ \varphi(\boldsymbol{x})) + \beta \, L_{\text{sup}}(\boldsymbol{w}^T \varphi(\boldsymbol{x}) + b, \, y_{\text{true}}). \tag{1}$$

For the supervised loss $L_{\text{sup}}$, we use the binary cross entropy. As unsupervised loss $L_{\text{un}}$, we use the commonly used standard flow loss obtained from the change of variables trick Dinh et al. (2016). The unsupervised loss ensures that inverting the function results in realistic looking images and can also be seen as a regularization.

The layer 342 used for the concept explanations is an affine coupling layer.

Table 7: Test performance of the supervised trained model MobileNet-V2 measured using a mean squared error (MSE).

| Attribute | Test MSE |
|---|---|
| Legs' Position | 0.0008912 |
| Bending | 0.0001915 |
| Background | 0.0001128 |
| Color | 0.0005297 |
| Rotation Pitch | 0.0009235 |
| Rotation Roll | 0.0005622 |
| Rotation Yaw | 0.002243 |
| Position X | 0.0004451 |
| Position Y | 0.0003912 |
| Shapes | 0.001102 |

## A.3 SUPERVISED MOBILENET-V2

We used a MobileNet-V2 to predict the attributes values. The models test mean squared errors are denoted in Table 7.

## B  LINKS TO MODEL, DATASET AND STUDY

Model export:
```
https://f002.backblazeb2.com/file/iclr2022/do_users_
benefit_from_interpretable_vision_model.tar.gz
```
Unbiased dataset:
```
https://f002.backblazeb2.com/file/iclr2022/two4two_
obj_color_and_spherical_finer_search_spherical_
uniform_0.33_uniform_0.15_unbiased.tar
```
Biased dataset:
```
https://f002.backblazeb2.com/file/iclr2022/two4two_
obj_color_and_spherical_finer_search_spherical_
uniform_0.33_uniform_0.15.tar
```
Export of study:
```
https://f002.backblazeb2.com/file/iclr2022/ICLR2022_
Export_Do_Users_Benefit_From_Interpretable_Vision.qsf
```
PDF print-out of the study:
```
https://f002.backblazeb2.com/file/iclr2022/ICLR2022_
Export_Do_Users_Benefit_From_Interpretable_Vision.pdf
```

## C  USER-STUDY LINKS AND VIDEOS

The studies can be accessed on the Qualtrics platform (with anonymized consent form) under the following links:

Baseline condition:
```
https://wznbm.qualtrics.com/jfe/form/SV_
7Umdmdaq8EHVRm6
```
INN condition:
```
https://wznbm.qualtrics.com/jfe/form/SV_
dneHADG7BxjVurc
```
Concepts condition:
```
https://wznbm.qualtrics.com/jfe/form/SV_
0PWErBQmGL0lobk
```

The tutorial videos can be viewed under the following links:

Introduction Tutorial for Peeky and Stretchy:
```
https://f002.backblazeb2.com/file/iclr2022/Intro_
Peeky_Stretchy.mp4
```
Second Introduction Tutorial for ML and Biases:
```
https://f002.backblazeb2.com/file/iclr2022/Second_
Intro_ML.mp4
```
Tutorial for baseline condition:
```
https://f002.backblazeb2.com/file/iclr2022/condition_
BASE.mp4
```
Tutorial for concept condition:
```
https://f002.backblazeb2.com/file/iclr2022/condition_
CONCEPTS.mp4
```
Tutorial for INN condition:
```
https://f002.backblazeb2.com/file/iclr2022/condition_
INN.mp4
```

## D  USER-STUDY PREREGISTRATION AND HYPOTHESIS

The Preregistrations can also be viewed under the following URLs:

- Validation of TWO4TWO: `https://aspredicted.org/blind.php?x=/62X_15J`
- Study 2: Concepts vs. Baseline and INN vs. Baseline: `https://aspredicted.org/blind.php?x=/7XN_77P`

We also paid the participants in both studies the more lucrative tariff included in the preregistration of study 2, e.g. for third comprehension task: 3.50GBP and so on.

### D.1  VALIDATION OF TWO4TWO

*1) Have any data been collected for this study already?*

No, no data have been collected for this study yet.

*2) What's the main question being asked or hypothesis being tested in this study?*

This study investigates whether users identify biases learned by a neural network. The neural networks task is to discriminate between two abstract animals ("Peeky" and "Stretchy"). Each participant is presented with predictions of the system in a 10x5 image grid.

After an initial tutorial phase, the participants have to find biases in the model. They do this by scoring different characteristics as relevant or irrelevant. The characteristics are: "legs position relative to the spine (LEGS)", "object color (COLOR)", "background (BACK)", "rounded or rectangular shape of the blocks (SHAPE)", and "rotation and bending (ROT)".

The main research question is whether we succeeded in creating a model that contains at least one bias that is hard to detect, i.e. either COLOR or SHAPE should be harder to detect than LEGS.

HB: Participants can identify the biases in COLOR or SHAPE less frequently than LEGS.

*3) Describe the key dependent variable(s) specifying how they will be measured.*

Participant will answer the following questions:

- LEGS: How relevant is the legs position relative to the spine for the system?: Relevant / Irrelevant
- COLOR: How relevant is the color of the animal for the system? Relevant / Irrelevant
- BACK: How relevant is the background of the animal for the system? Relevant / Irrelevant
- SHAPE: How relevant is the rounded or rectangular shape of the animal's blocks for the system? Relevant / Irrelevant
- ROT: How relevant is the rotation and bending of the animal for the system? Relevant / Irrelevant

The ground truth answer is that LEGS, COLOR, SHAPE are relevant while BACK and ROT are irrelevant. Our first dependent variable is the number of times the head position was selected as relevant. Our second dependent variable is the number of times the color of the animal was selected as relevant. Our third dependent variable is the number of times the rounded or rectangular shape of the animal's blocks was selected as relevant.

*4) How many and which conditions will participants be assigned to?*

Our study follows a within-subject design and has only one condition. We first show the participants introductory videos about the two abstract animals, the machine learning system, and some guidance on how to interpret the predictions of the system. Each video is accompanied by a written summary. We then show the predictions of the system in a grid of images: 10 sorted rows of 5 images drawn from the validation set (50 original images). Each of the five columns represents the neural netwoks's logit range. Similarly rated images are assigned to the same column.

*5) Specify exactly which analyses you will conduct to examine the main question/hypothesis.*

We will conduct two exact one-sided McNemar-tests with LEGS acting as our control: one between SHAPE and LEGS and a second between COLOR and LEGS. We will use a one-sided test as we expect that SHAPE and COLOR are harder to identify. The significance level of both tests will be Bonferroni adjusted to $\alpha = 0.025$.

*6) Describe exactly how outliers will be defined and handled, and your precise rule(s) for excluding observations.*

We reject participants with low effort responses or who failed to understand the dataset, machine learning concept, or explanation method. We have implemented hard-coded exclusion criteria directly in the survey (implemented with Qulatrics and Prolific).

- did not finish experiment at all or in under 77 minutes
- did not watch tutorial videos completely (there are 3 videos) or failed a multiple-choice comprehension test twice (there are four such tests), unless participants explicitly ask us to retake the study
- using a device smaller than a tablet (min. 600 px in width or height)
- provided answers about relevant characteristics in under 30 seconds
- withdrawn data consent / returned task on Prolific
- circumvented Qualtrics protection against retaking the entire survey again (first complete submission will be counted)

We do not plan to exclude any participants who passed all of the above criteria unless the qualitative answers reveal a serious misunderstanding of the study instructions that the multiple choice tests did not cover. We will report such exclusions in detail in the Appendix.

*7) How many observations will be collected or what will determine sample size?*

No need to justify decision, but be precise about exactly how the number will be determined. 50 participants from Prolific with the background:

- Fluent in English
- Hold an academic degree
- Prolific approval rate of at least 90%
- Did not participate in pilot studies
- Passed hard coded exclusion criteria (see 8).

We pay participants max. 8.00 GBP (6.00 GBP base salary + 2.00 GBP max bonus). For those failing any comprehension questions or not watching the video, we pay:

- First comprehension task: no compensation
- Second comprehension task: 0.5 GBP
- Third comprehension task: 1.75 GBP
- Failed to watch first video: no compensation
- Failed to watch second video: 1 GBP
- Failed to watch third video: 2 GBP

*8) Anything else you would like to pre-register?* (e.g., secondary analyses, variables collected for exploratory purposes, unusual analyses planned?) We ask participants to answer three multiple choice comprehension tests in the form of true/false statements to ensure that they understood the task and the dataset. We also ask them to provide some free-text justification of why they chose a relevant / irrelevant rating to the questions in Section 3.

### D.2 STUDY 2: CONCEPTS VS. BASELINE AND INN VS. BASELINE

*1) Have any data been collected for this study already?*

No, no data have been collected for this study yet.

*2) What's the main question being asked or hypothesis being tested in this study?*

This study investigates whether users identify biases learned by a neural network. The neural networks task is to discriminate between two abstract animals ("Peeky" and "Stretchy"). Each participant is presented one of three different explanation methods: baseline (B), counterfactuals obtained using invertible neural networks (CF) and prototypes (P).

Each participant is randomly assigned to a method. After an initial tutorial phase, the participants have to find biases in the model. They do this by scoring different characteristics as relevant or irrelevant. The characteristics are: "legs position relative to the spine (LEGS)", "object color (COLOR)", "background (BACK)", "rounded or rectangular shape of the blocks (SHAPE)", and "rotation and bending (ROT)".

The main question of our study is whether the participants can correctly identify relevant and irrelevant attributes using these explanation methods (B, CF, P). This is reflected by two hypotheses:

H1: *Participants identify relevant and irrelevant attributes with less accuracy using P compared to B.*

H2: *Participants identify relevant and irrelevant attributes with higher accuracy using CF compared to B.*

*3) Describe the key dependent variable(s) specifying how they will be measured.*

Participant will answer the following questions:

- LEGS: How relevant is the legs position relative to the spine for the system?: Relevant / Irrelevant
- COLOR: How relevant is the color of the animal for the system? Relevant / Irrelevant
- BACK: How relevant is the background of the animal for the system? Relevant / Irrelevant
- SHAPE: How relevant is the rounded or rectangular shape of the animal's blocks for the system? Relevant / Irrelevant
- ROT: How relevant is the rotation and bending of the animal for the system? Relevant / Irrelevant

The ground truth answer is that LEGS, COLOR, SHAPE are relevant while BACK and ROT are irrelevant. Our dependent variable is the percentage of correctly answered questions per participant (accuracy, which is computed as (true positives + true negatives)/number of total answers).

*4) How many and which conditions will participants be assigned to?*

We run a between-subject study, with randomly but equally assigned participants to 1 of 3 conditions. We first show introductory videos about the two abstract animals, the machine learning system, the explanation technique and some guidance on how to interpret the technique. Each video is accompanied by a written summary. We then show a grid of (10x5) images:

1. B: NN predictions explained with 10 sorted rows of 5 images drawn from the validation set (50 original images). Each of the five columns represents a score range. Similarly rated images are assigned to the same column.

2.CF: Same grid layout as B, but the NN is explained by counterfactual interpolations. Each row contains interpolations which change the prediction of the NN to fit the designated score. Original images are used as starting points but are not shown.

3.P: We found concepts based on the work by (Zhang et al., 2020). Each row shows a set of relevant concepts. We only used concepts correlated with at least r=0.2 with the model logit values. In total, we display 10 rows where each row contains a concept. Each row contains a set of 5 example images for which the concept is relevant.

(Zhang et al., 2020) https://arxiv.org/abs/2006.15417

*5) Specify exactly which analyses you will conduct to examine the main question/hypothesis.*

We will compute the accuracy scores for each participant and then compare the accuracy scores between the conditions. We expect the data to be non-normally distributed, and will test this assumption using a Shapiro-Wilk test with a significance level of $\alpha = 0.05$. If our assumption is true, we plan to conduct a Kruskal-Wallis test, followed by post-hoc analysis using Wilcoxon's-rank-sum tests for focused comparison between the groups CF and B (expecting higher accuracy in CF) and P and B (expecting lower accuracy in P).

If the data is normally distributed, we will conduct a one-way ANOVA with planned contrasts, if the following assumptions of ANOVAs are met:

- Homogeneity of the variance of the population (assessed with a Levene-Test with a significance level of $\alpha = 0.05$.)

If the homogeneity of variance assumption of ANOVA is violated (assessed with a Levene-Test with a significance level of $\alpha = 0.05$.), we plan to perform Welch's Anova.

*6) Describe exactly how outliers will be defined and handled, and your precise rule(s) for excluding observations.*

We reject participants with low effort responses or who failed to understand the dataset, machine learning concept, or explanation method. We have implemented hard-coded exclusion criteria directly in the survey (implemented with Qulatrics and Prolific).

- did not finish experiment at all or in under 77 minutes
- did not watch the tutorial videos completely (there are 3 videos) or failed a multiple-choice comprehension test twice (there are four such tests), unless participants explicitly ask us to retake the study
- using a device smaller than a tablet (min. 600 px in width or height)
- provided answers about relevant characteristics in under 30 seconds
- withdrawn data consent / returned task on Prolific
- circumvented Qualtrics protection against retaking the entire survey again (first complete submission will be counted)

We do not plan to exclude any participants who passed all of the above criteria unless the qualitative answers reveal a serious misunderstanding of the study instructions that the multiple choice tests did not cover. We will report such exclusions in detail in the Appendix.

*7) How many observations will be collected or what will determine sample size? No need to justify decision, but be precise about exactly how the number will be determined.*

240 (80 per condition) participants from Prolific with the background:

- Fluent in English
- First, we sample participants with an academic degree. If we do not reach the desired participant number, which is likely given the limited availability of such subjects, we will supplement with participants with an academic degree in other subjects. All participants will be randomly and equally split into the 4 conditions.
- Prolific approval rate of at least 90%
- Did not participate in pilot studies
- Passed hard coded exclusion criteria (see 8).

We pay participants max. 6.50 GBP (4.50 GBP base salary + 2.00 GBP max bonus). For those failing any comprehension questions or not watching the video, we pay:

- First comprehension task: 0.5 GBP
- Second comprehension task: 1.75 GBP

- Third comprehension task: 3.50GBP
- Failed to watch first video: no compensation
- Failed to watch second video: 1 GBP
- Failed to watch third video: 2 GBP

*8) Anything else you would like to pre-register? (e.g., secondary analyses, variables collected for exploratory purposes, unusual analyses planned?)*

In a previous study, we collected 50 responses for the baseline condition only (Preregistration #75056)). We do not plan to use the data for this study.

We ask participants to answer three multiple choice comprehension tests in the form of true/false statements to ensure that they understood the task and the dataset. We also ask them to provide some free-text justification of why they chose a relevant / irrelevant rating to the questions in Section 3.

Additionally, we ask the participants about their machine learning expertise level. Participants can rate their expertise as: complete novice, some expertise, or expert in the topic. We plan to use descriptive statistics to see how accuracies change per condition for each expertise level and how expertise was distributed within our sample.

We are also planning a qualitative thematic analysis of the open-text questions in our survey via open and axial coding, with the aim of understanding how participants integrated explanations in their reasoning about the relevance of attributes.

