# OpenReview forum: "Do Users Benefit From Interpretable Vision? A User Study, Baseline, And Dataset"
_ICLR.cc/2022/Conference — ICLR 2022 Poster_

### Official Review · Reviewer_gzkV · 2021-10-26

**Correctness:** 4
**Technical Novelty And Significance:** 3
**Empirical Novelty And Significance:** 3
**Recommendation:** 8
**Confidence:** 3

**Main Review:**

I really liked this paper! It's refreshing in this community to read
work that focuses on users, rather than models. Evaluating
interpretability methods is indeed difficult, and the authors clearly
did a significant amount of design, pre-registration, user
coordination, etc.  in service of their goal. After watching the
training videos myself (linked in the appendix), I found that the
authors conclusion interesting, but unsurprising --- in this
controlled scenario, pixel highlighting (as in concept explanability)
is ambiguous because many different attributes may show up in single
highlight.

While I was generally appreciative and positive about the work, it
does have some shortcomings.

1. The stretchy vs. peeky environment, while cute, is quite different
   than most computer vision tasks. While I understand the motiviation
   for being able to generate images with particular properties, I do
   wonder if the results here would transfer to, e.g., object
   classification in photographic images.

2. The stretchy vs. peeky environment is a binary classification
   setup, which gives rise to the baseline method of simply showing
   predictions that the authors compare against. However, for more
   than two classes, this baseline is not directly usable. If there
   were 1K classes, it wouldn't be possible to lay out model
   predictions in a 2D grid. My suspicion is that this is partly a
   motivating factor for a method like concept highlighting where
   pixels are highlighted in input space. For higher dimensional
   classification tasks, it may be impractical or impossible to show a
   2D grid for correct vs. incorrect for all classes; so, something
   like pixel highlighting might be more useful in that case.

**Summary Of The Paper:**

The authors explore several explanation methods for image classifiers
via a user study. They study a toy environment containing images of
two animals: stretchy (who has stretched legs) and peeky (who has a
head that extends beyond its front legs). While the true label of
peeky vs. stretchy is defined deterministically and specifically, the
toy environment enables the authors to generate spurrious correlations
between attributes like background color, animal position, shape,
etc. and the label (which the model picks up on). The goal of the
explanation methods is to help users identify which features
(spurrious and not) the model is picking up on. Through a series of
pre-registered user studies comparing a simple baseline to
counterfactual explanations to concept highlighting, they explore
whether or not users can accurately reconstruct which features the
model is using in its predicitons. They find that concept highlighting
performs far worse than the baseline of simply showing some model
predictions in a grid, and that the baseline and the counterfactual
method performed similarly. They release their procedures and
generation code as a challenge to the community: can a new
interpretability method outperform their baseline?

**Summary Of The Review:**

Overall, I think more work like this should appear at ICLR ---
explanability is an important topic in machine learning, and, while a
user study is perhaps less conventionally presented at ICLR, I think
it should be. Furthermore, I admire the experimental practices
presented here, e.g., preregistration. I think my two big critiques
(toy-ness not applying to more common image clf problems, and
non-binary tasks) can be addressed via presentation updates in
revision.

---

> ### Author Response · Authors · 2021-11-13
> **Answer to gzkV**
>
> Thank you for your positive assessment. We agree that more user studies
> should appear at ICLR.
>
> > The stretchy vs. peeky environment, while cute, is quite different
> > than most computer vision tasks. While I understand the motiviation for
> > being able to generate images with particular properties, I do wonder if
> > the results here would transfer to, e.g., object classification in
> > photographic images.
>
> We agree with the reviewer that results may differ on natural image
> datasets. However, this is a general problem: results may not generalize
> from any dataset to another for multiple reasons: participants might
> have more or less helpful prior knowledge, the model performance might
> differ, or the dataset features are easier/harder to understand.
>
> When we created Two4Two, our objective was to translate challenges faced
> on "real" computer vision data (like spatially overlapping features)
> into an abstract domain where at least some of these confounding factors
> can be controlled. Of course some properties of photorealistic datasets
> are lost in our abstraction. However, we gain a lot of methodological
> validity in return: we can control feature relevance and have ground-truth
> information for it. No other dataset for human subject evaluation
> offers such fine-grained control, not even datasets that introduce very
> obvious biases in natural image datasets, which are the closest
> alternative to our dataset. For example, the BAM dataset pastes segmented
> objects onto certain backgrounds, which is a very obvious bias and does
> not look *real* at all. Still, it suffers from the same problems of
> generalizability as any other dataset.
>
> > The stretchy vs. peeky environment is a binary classification setup,
> > which gives rise to the baseline method of simply showing predictions
> > that the authors compare against. However, for more than two classes,
> > this baseline is not directly usable. If there were 1K classes, it
> > wouldn\'t be possible to lay out model predictions in a 2D grid.
>
> Your criticism is valid. This should be mentioned as a limitation and
> potential for future work in the paper. The baseline is particularly
> good in the 1-vs-1 class setup. However, even in multi-class problems,
> the information can be rearranged to show contrastive explanations (Why
> one class and not the other - [Miller et al.,
> 2017](https://arxiv.org/abs/1706.07269)), e.g. by contrasting the
> highest scored class with the second highest. However this is only a
> feasible explanation method for a limited number of relevant classes.
> Explaining image classifications locally for classification problems
> where a larger number of classes exist and are relevant for the user is
> an understudied phenomenon in general: We are not aware of any user
> study where participants had to reason about more than just a few
> classes. We, therefore, focused on the simple scenario of binary
> classification. In future work, it should be investigated how the
> baseline performs when multiple classes are relevant.
>
> > My suspicion is that this is partly a motivating factor for a method
> > like concept highlighting where pixels are highlighted in input space.
> > For higher dimensional classification tasks, it may be impractical or
> > impossible to show a 2D grid for correct vs. incorrect for all classes;
> > so, something like pixel highlighting might be more useful in that case.
>
> We believe that the usefulness of highlighting pixels depends more on
> the abstraction of the datasets. The more abstract it is, the less
> sensible it is to highlight pixels.
>
> If you have any additional questions, we are happy to clarify them.

---

> > ### Comment · Reviewer_gzkV · 2021-11-14
> > **Response to authors**
> >
> > Thanks for your response! I think we're on the same page with most things, and I'm (clearly) positive on your work! The only part I would push back on is generalizability. While, yes, any two datasets will have some disconnect between them, the contexts in which people generally use interpretability methods like this (photographs of everyday objects) significantly differ from Two4Two. So, while I totally respect and understand the need for a controlled environment (and value that as a contribution), it's still not entirely clear how these results would translate to more commonly considered visual domains/tasks.

---

> > > ### Author Response · Authors · 2021-11-23
> > > **Response to reviewer gzkV**
> > >
> > > Thank you for your answer. We agree that there is a difference between natural datasets and Two4Two. We add a paragraph to the conclusion in the updated manuscript:
> > >
> > > > We believe that the results also have implications for natural image data.
> > > When we created Two4Two, our objective was to translate challenges faced on "real" computer vision data (like spatially overlapping features) into an abstract domain. Although some properties of photorealistic datasets are lost in this abstraction,
> > > a method performing poorly on Two4Two would likely not perform well on a natural dataset with spatially overlapping features.

---

### Official Review · Reviewer_3sBP · 2021-10-29

**Correctness:** 3
**Technical Novelty And Significance:** 2
**Empirical Novelty And Significance:** 3
**Recommendation:** 5
**Confidence:** 5

**Main Review:**

Strength:
-------------
(1) This paper is well written, and it provides a nice method for establishing an artificial dataset where the ground-truth explanation is available. It also reveals the problem that many model explanation methods encounter: is the explanation useful for users?
(2) The user study includes many participants, which provides a solid result.

Weakness:
---------------
(1)	Is it fair to compare concept model in this dataset/ user study? First, the attributes (biases) that should be understood by users are very abstract and cannot easily visualized in concept activation maps. Second, if the users see only 5 images from one concept class (as shown in Fig.1 c), it is not clear which attribute is important for the decision. However, on other two conditions (B and CF), they have the classes and confidence bar above. Does the concept model also have this when shown to the users?

(2)	Could the participant in study 1 also participant in study 2? If so, the answers may get interfered since the users might already know which attributes were important and can bring that knowledge into their answers.

(3)	How did the classifier used for computing explanations as well as in baseline perform? It may also influence the quality of explanations generated.

(4)	The main concern is whether the proposed dataset is general enough to fairly compare different explanation methods on it. For instance, there is a large number of attribution-based explanations. They are visually similar as concept-based maps, which might also fail in highlighting the attribute such as shape or color.

(5)	In Table 2, what do the stars imply?


**Summary Of The Paper:**

This paper proposes a dataset called TWO4TWO to conduct a user-study on two interpretability methods: counterfactual and concept-based explanations. Since the dataset is generated and can be fully controlled by users, the ground-truth important attributes to the decision are known. In this case, we know the ground-truth features and can thus assess whether the model explanation also reveal the correct features. As a baseline explanation, the authors group the input according to the model’s output logits. The result from the user study shows that the two sophisticated explanation models don’t surpass the simple baseline method, which indicates that explanation techniques shall be evaluate in user studies.

**Summary Of The Review:**

The authors reveal an interesting research question and provide with a user study design. The results from the user study seem solid. However, the dataset and the questions are not general and proper to assess concept-based or attribution-based explanations. The authors should first improve or clarify this issue. Otherwise, the dataset may not be very useful.

---

> ### Author Response · Authors · 2021-11-13
> **Answer Reviewer 3sBP Part 2**
>
>
> > (4) The main concern is whether the proposed dataset is general
> > enough to fairly compare different explanation methods on it. For
> > instance, there is a large number of attribution-based explanations.
> > They are visually similar as concept-based maps, which might also fail
> > in highlighting the attribute such as shape or color.
>
> Two4Two was designed as a challenging dataset to evaluate explanation
> methods technically and in human-subject experiments. It was not
> designed to showcase the abilities and advantages of existing methods
> but instead to draw attention to unsolved challenges.
>
> For example, overlapping input features
> are very common in histology
> cuts: both the cell's shape and texture can be important. Human faces
> also naturally have many overlapping features, e.g. smiling affects many
> facial regions. Attribution based methods struggle with such spatially
> overlapping features. A limitation pointed out by prior work (Alqaraawi
> et al., 2020). A further general limitation is that feature relevance
> can be difficult to explain by highlighting regions in an image, a
> limitation that became evident in our experiment for concept maps as
> well (N=59). This is not only the case for spatially overlapping
> features but for any feature not localizable to small pixel regions
> (e.g. overall brightness of image).
>
> > (5) In Table 2, what do the stars imply?
>
> The stars indicate statistically significant differences to the baseline
> condition. We will update the table's caption.
>
> > The authors reveal an interesting research question and provide with
> > a user study design. The results from the user study seem solid.
> > However, the dataset and the questions are not general and proper to
> > assess concept-based or attribution-based explanations. The authors
> > should first improve or clarify this issue. Otherwise, the dataset may
> > not be very useful.
>
> As stated before, our dataset was designed as a challenge for current
> explanation methods and includes spatially overlapping features. This
> also occurs in natural datasets and might have ethical implications
> (e.g. for face data, race and gender represent spatially overlapping
> features; it would not be possible to tell from a highlighted face if it
> was highlighted because of its gender or race). But even if one
> considers spatially overlapping relevant features as unfair for
> evaluating attribution based methods this does not affect our
> contribution of Two4Two as it is capable of generating data without
> spatially overlapping relevant features (e.g. bias on object color and
> background color).
>
> We believe that empirical studies like ours are helpful to guide
> technical advances in interpretable ML. Otherwise, we might risk
> developing techniques not useful to humans. Unfortunately, pre-registered user studies
> that evaluate interpretable deep vision are rare to find, a study design
> based on the bias discovery task is not yet established, nor a required
> data generator with ground-truth feature importances. This year's ICLR
> aims to strengthen such empirical contributions.

---

> ### Author Response · Authors · 2021-11-13
> **Answer Reviewer 3sBP Part 1**
>
> Thank you for taking your time to review our manuscript. We
> appreciate that you found our paper well written, liked the dataset with
> ground-truth feature importances and found the results solid.
>
> > (1) Is it fair to compare concept model in this dataset/ user study?
> > First, the attributes (biases) that should be understood by users are
> > very abstract and cannot easily visualized in concept activation maps.
> > Second, if the users see only 5 images from one concept class (as shown
> > in Fig.1 c), it is not clear which attribute is important for the
> > decision. However, on other two conditions (B and CF), they have the
> > classes and confidence bar above. Does the concept model also have this
> > when shown to the users?
>
> In [(Zhang et al.,
> 2020)](https://arxiv.org/pdf/2006.15417.pdf)
> it is claimed:
>
> > "Our framework provides both local and global concept-level
> > explanations for pre-trained CNN models."
>
> Since concepts can explain feature relevance globally and contain
> multiple relevant features at the same time, this explanation technique
> has the potential to outperform all other methods.
>
> In practice, however, this method seems to suffer from technical
> limitations. The automatically discovered concepts were not strongly
> correlated with the output class (r is in the range \[0.21, 0.34\]).
> Suppose we would have contrasted a concept along the dimension "more
> peeky" – "more stretchy", then the "more peeky/stretchy" column would
> also include objects of the opposite class due to the low class
> correlation. Participants would have been confused when a "more peeky"
> column includes Stretchies.
>
> We could have simply omitted the Stretchies for the "more peeky" column,
> but this would suggest that the concept does not appear for Stretchies
> – a faulty explanation.
>
> Furthermore, users were only asked *if* a feature is relevant and not
> *how* relevant a feature is for one particular class. Hence, if concepts
> would have highlighted relevant features accurately, no additional
> confidence information would have been needed.
>
> We used five images per concept. This number was also used in [(Zhang
> et al.,
> 2020)](https://arxiv.org/pdf/2006.15417.pdf).
> This work showed that five images are enough to successfully sort images to
> the matching concepts. We also experimented with more images, but did not find
> them to present a concept any clearer. The main problem remains that
> automatically-discovered concepts do not contain a semantic meaning.
>
> In the updated version of our manuscript, we will add some more details
> on the presentation of the concepts and also discuss this limitation of
> the concept evaluation.
>
> > (2) Could the participant in study 1 also participant in study 2? If
> > so, the answers may get interfered since the users might already know
> > which attributes were important and can bring that knowledge into their
> > answers.
>
> Participants could not take part in both studies. We also excluded
> participants from any pre-studies. We will mention this in the updated
> version of the paper.
>
> > (3) How did the classifier used for computing explanations as well
> > as in baseline perform? It may also influence the quality of
> > explanations generated.
>
> Thank you for pointing this out. The model had a test accuracy of 96.7%.
> We will include these numbers in the updated version of our paper.

---

> ### Author Response · Authors · 2021-11-29
> **Please answer to our rebuttal**
>
> Dear 3sBP,
>
> we posted our initial answers more than two weeks ago and have updated the manuscript last week. However, you did not answer our rebuttal. Can you please address our rebuttal and provide a justification for your score? We submitted a valid and faithful manuscript and expect a review process that evaluates our work fairly.
>
> Best,
> the Authors

---

### Official Review · Reviewer_pBBz · 2021-10-30

**Correctness:** 3
**Technical Novelty And Significance:** 2
**Empirical Novelty And Significance:** 3
**Recommendation:** 5
**Confidence:** 5

**Main Review:**

For the proposed dataset, more examples are expected given for each of the two classes. The definition of ‘legs moved inwards’ is not rigorous or clear. The same for ‘legs stretched out’.  For example, for Peeky, does it mean as long as one pair of legs moving to the center of the body? Or both have to. How to determine which two nodes belong to a pair. Why not the upper left and upper right. Also, the definition of ‘legs moved inwards’ should be defined like angle or something like that more rigorously in math.

What is the main feature? The four ‘body posture (bending and three rotation angles), position, animal color (from red to blue), blocks’ shape (from cubes to spheres), and background color (from red to blue)?’. Are they continuous or discrete? For this sentence,
‘A single scalar encodes the legs’ position.’, is ‘legs’ position’ is a feature? How to measure it? Or is there any visualization showing what 0 or 1 look like?

I’m a little confused with this statement. ‘we sampled the block’s shape in a non-predictive biased fashion. For very stretched legs’ positions (Stretchy), the data distribution was biased towards round blocks, while for more retracted legs’ positions (Peeky), most blocks were cubic.’ If two classes are sampled from different distributions, shouldn’t it be predictive?

For ‘We generated the concepts using layer 342 (from a total of 641).’, what is layer 342?

For the user study, what task the online workers are asked to do. What’s the question to ask so that they are asked to identify the class-relevant features. Or is there an interface example?

Are there any statistics for the user study, like time spent? Maybe this can reflect if the task is hard, an evidence for the argument in section 5.1.

The first line under section 5.3, ‘As stated iWe automatically’. iWe? Just a typo?

One paper that uses a similar synthetic dataset. Maybe useful for the concerned task in this paper.
Yuxin Chen, Oisin Mac Aodha, Shihan Su, Pietro Perona, Yisong Yue’ Near-Optimal Machine Teaching via Explanatory Teaching Sets’

**Summary Of The Paper:**

The paper proposed a synthetic dataset to explore the bias contained in the dataset. Because the dataset is synthetic. We can manually change its attributes, add or eliminate the bias. Then, two main user study is conducted. One is to see if users can find the bias and another is to investigate if explanations are helpful.

**Summary Of The Review:**

My major concern is that this paper does not have any technical contributions. And the proposed dataset is not very interesting or distinct compared to the existing. The dataset building or description is also not very rigorous.

---

> ### Author Response · Authors · 2021-11-13
> **Answer to pBBz Part 2**
>
> > Are there any statistics for the user study, like time spent? Maybe
> > this can reflect if the task is hard, an evidence for the argument in
> > section 5.1.
>
> For an online study, time might not reflect the difficulty of the task
> well. Since the experiment is taken in a less controlled environment
> than a lab, users might interrupt the task or become distracted
> (children, telephone calls, getting something to drink or eat).
> Additionally, timing also reflects effort. Some participants might be
> inclined to try very hard, while others may just rush through the task.
> While we assume that high and low effort participants will be equally
> distributed among conditions and their influence on performance data is
> negligible, it invalidates timing data due to much higher
> variance. Furthermore, we ask users to provide full sentence
> justifications as part of the task. Thus, the time spent is highly
> influenced by their language skill and crowd-working experience.
>
> We calculated the timings, and as expected the standard deviation is
> high:
>
> ```
> cond mean sd min max
>
> 1 INN 671.2907 305.5477 187.281 1412.226
>
> 2 CON 757.5362 455.1977 215.991 2389.288
>
> 3 BASE 796.2906 463.4102 194.772 2357.933
> ```
>
> A t-test on the differences between the conditions
> does not
> reveal a statistical significance (INN vs.
> BASE: p=0.065; INN vs. CON: p=0.226; CON vs. BASE: p=0.632;
> all p-values are uncorrected for multiple comparison). As we argued
> above, even if the results would be more conclusive, inferring the
> hardness of the task from the timing can be misleading.
>
> > One paper that uses a similar synthetic dataset. Maybe useful for the
> > concerned task in this paper. Yuxin Chen, Oisin Mac Aodha, Shihan Su,
> > Pietro Perona, Yisong Yue' Near-Optimal Machine Teaching via Explanatory
> > Teaching Sets. \[\...\] And the proposed dataset is not very interesting
> > or distinct compared to the existing.
>
> Thank you for pointing us to this work. We were not aware of it and will
> include it in the related work section. The Two4Two dataset offers
> several advantages over this work:
>
> -   Two4Two is specifically designed for human-subject experiments, and
>     we open source all study material, including the dataset, while no
>     source code is available for (Chen et al., 2018).
>
> -   Two4Two allows the creation of overlapping classes, e.g. arm
>     position in the center, whereas for (Chen et al., 2018) the class
>     can be always determined exactly, e.g. {blue, gray square, thick} →
>     Mars. Adding a bias to this dataset might therefore be difficult
>     because the class can be predicted exactly.
> -   Two4Two allows much more control over the data generation process.
>     Six continuous variables can be manipulated and arbitrarily correlated with one
>     another. This allows for the creation of
>     obvious, and difficult biases – a claim we verified in the
>     first user study.
>
> > My major concern is that this paper does not have any technical
> > contributions.
>
> The ICLR Program Chairs sent an e-mail on 10/3/2021 which stated:
>
> > This year, we have made changes to the review forms: reviewers are
> > asked to assess correctness, technical novelty and significance
> > (algorithms, models, and theories), as well as empirical novelty and
> > significance (advancements, insights, or datasets) separately.
> > Submissions with significant contributions in either technical aspects
> > or empirical aspects will be given high priority for acceptance.
>
> Unfortunately, this was only communicated via email and not included in
> the call-for-papers. It might even be that this email was sent only to
> the authors and not the reviewers. We assume that you were not aware of
> this recent change.
>
> A technical contribution of our work is the ground-truth measuring of
> feature importances. However, the main focus of our work are the
> empirical contributions: the dataset, baseline, experimental design, and
> study results. It is also not uncommon to publish user studies at ICLR,
> e.g. [(Borowski et al.,
> 2021)](https://openreview.net/forum?id=QO9-y8also-)
> from last year.
>
> We believe that empirical studies like ours are helpful to guide
> technical advances. Furthermore, pre-registered user studies that
> evaluate interpretable ML methods are rare to find in our community, a
> study design based on the bias discovery the task is not yet
> established, nor a required data generator with ground-truth feature
> importances.
>
> We would kindly ask you to update your review and also evaluate our
> empirical contributions (dataset, study design, baseline, open sourcing
> our assets).

---

> ### Author Response · Authors · 2021-11-13
> **Answer to pBBz Part 1**
>
> Thank you for taking the time to review our paper.
>
> > For the proposed dataset, more examples are expected given for each
> > of the two classes. The definition of 'legs moved inwards' is not
> > rigorous or clear. The same for 'legs stretched out'. For example, for
> > Peeky, does it mean as long as one pair of legs moving to the center of
> > the body? Or both have to. How to determine which two nodes belong to a
> > pair. Why not the upper left and upper right. Also, the definition of
> > 'legs moved inwards' should be defined like angle or something like that
> > more rigorously in math.
>
> We thank you for raising this concern. We will include a clearer and
> more rigorous description in the updated manuscript:
>
> The main property is the legs\' position relative to the spine. For both
> animals, one pair of legs is always at a constant position at one end of
> the spine. The other end is manipulated by the attribute "leg's
> postion". It moves the legs parallel to the spine. It is defined as a
> scalar \[0,1\]. At a value of 0.5,the pair of legs are at the same
> vertical position as the last block of the spine. We wanted to create
> some ambiguity to ensure a model has an incentive to use possible
> biases. Therefore, the leg's position is not predictive between 0.48 and
> 0.52. At that range, the legs are neither moved inwards nor outwards.
> Peekies have a leg position of x \< 0.52 which means legs are moved
> inwards to the body center. The scalar value of 0 represents the most
> extreme case of legs being moved inwards. In the same fashion,
> Stretchies are extended outwards, meaning x \> 0.48, where 1
> represents the most extreme case of legs being moved outwards (leg\'s
> position = 1).
>
> Figure 2 illustrates the difference visually but we plan to include more
> visual examples of the different leg positions in the Appendix.
>
> We would also like to point out that we took extensive measures to
> ensure that participants understood the difference between the two
> animals without such a formal definition. We encourage you to take a
> look at the short video instruction:
> https://f002.backblazeb2.com/file/iclr2022/Intro_Peeky_Stretchy.mp4
>
> After watching this video, participants had to identify four animals
> correctly to commence the study.
>
> > I'm a little confused with this statement. 'we sampled the block's
> > shape in a non-predictive biased fashion.
>
> Thank you for pointing out that this sentence lacks clarity. The main
> point is that the blocks' shape does not contain any additional
> class-discriminative information which is not already included in the
> leg's position. We changed the sentence to: "we sampled the block\'s
> shape in a biased fashion. However it does not contain any additional
> class-discriminative information not already encoded in the leg's
> position.".
>
> > For 'We generated the concepts using layer 342 (from a total of
> > 641).', what is layer 342?
>
> Layer 342 is an affine coupling layer. We did not specify this in
> the main text as this kind of layer is specific to invertible neural
> networks for vision. We do not explain the model architecture in the
> main text so as not to distract the reader. We will update the Appendix
> and state there the type of layer 342.
>
> > For the user study, what task the online workers are asked to do.
> > What's the question to ask so that they are asked to identify the
> > class-relevant features. Or is there an interface example?
>
> Section 5.2 explains that "For both studies, after the tutorial phase,
> participants were shown the explanations and then asked to assess the
> relevance of the five attributes: legs' position relative to the spine,
> animal color, background, rotation or bending, and blocks' shape. The
> questions were formulated as:"How relevant is\< attribute \>for the
> system?", and participants had to choose between irrelevant or relevant.
>
> We plan to update the section "Manageable but not Oversimplified Tasks"
> to begin with a clear definition of the task: "We propose the task of
> rating feature relevance, which directly reflects users\' perception on
> feature importance. Participants are asked to rate features as either
> relevant or irrelevant to a model."
>
> In addition, we now provide also a PDF print-out of the study design:
> https://f002.backblazeb2.com/file/iclr2022/ICLR2022_Export_Do_Users_Benefit_From_Interpretable_Vision.pdf

---

> ### Author Response · Authors · 2021-11-29
> **Please answer to our rebuttal**
>
> Dear pBBz,
>
> we posted our initial answers more than two weeks ago and have updated the manuscript last week. You increased your rating to 5, but did not provide any justification. Can you please summarize how you justified your score? We submitted a valid and faithful manuscript and expect a review process that evaluates our work fairly.
>
> Best,
> the Authors

---

### Official Review · Reviewer_rVJ7 · 2021-11-02

**Correctness:** 4
**Technical Novelty And Significance:** 3
**Empirical Novelty And Significance:** 4
**Recommendation:** 6
**Confidence:** 3

**Main Review:**

Strength:
1. This is a very novel and clever experiment design that has a clear definite ground truth solution and irrelevant other features that may confuse the ML systems. Those other features are visually overlapping (shape, color, etc) and are hard to be visually explained.
2. The results are interesting that under these extreme experiments, the currently popular explanation approach can not outperform the confidence represented by the logits.

Weakness:
Besides the limitation stated on the last page:
1. The experiment protocol requires a controllable generator of the training data which is hard to extend to other visual tasks (for example image manipulations). It is hard to generate data are on the manifold of the natural data for using the current approach to test the explanation results for real data

**Summary Of The Paper:**

The paper introduces an experiment design and an approach to synthesizing the dataset for the experiment. The experiment asked the systems to classify whether the shape of the "animal" is Peaky or Stretchy. The advantages are that authors can controllably generate trainable examples under arbitrary biases of the predefined features (shape, color, etc). For experiments, human subjects are asked to predict the systems' output. The authors compared the visual explanation (concept explanation) and counter-factual explanation with the baseline explanation that uses the output logits and found that the the current good explanation approaches do not make humans understand the system significantly better

**Summary Of The Review:**

The paper provides a  novel and clever experiment design to show that the limit of the current explanation approaches but shows limited potential to be used as a good protocol for large-scale real data tasks.

---

> ### Author Response · Authors · 2021-11-13
> **Answer to Reviewer rVJ7**
>
> Thank you for taking your time to review our manuscript. We appreciate
> that you found our experimental design novel and clever and that you
> scored correctness and empirical contributions with 4.
>
> > The experiment protocol requires a controllable generator of the
> > training data which is hard to extend to other visual tasks, \[\...\].
> > It is hard to generate data are on the manifold of the natural data \--
> > Reviewer rVJ7
>
> We want to emphasize that a user study on bias discovery is impossible
> with natural datasets: a natural dataset might introduce an unknown
> bias, we cannot estimate ground-truth feature importances, and
> participants might have their own preconceptions about the dataset.
>
> We disagree that it is hard to generate natural data. Researchers have
> demonstrated that it is possible to create photorealistic
> training data artificially ([Alhaija et al., 2018](
> https://link.springer.com/content/pdf/10.1007/s11263-018-1070-x.pdf))
> Other work builds on game engines (see Table here for an overview
> https://apollo.auto/synthetic.html). In both cases, we could include
> interventions in the data generation process. This is not a limitation
> of the experiment protocol.
>
> > The results are interesting that under these extreme experiments, the
> > currently popular explanation approach can not outperform the confidence
> > represented by the logits.
>
> Thank you for finding our results interesting. However, we disagree that
> the user task was \"extreme\" or "hard\". Spatially overlapping features
> are widespread, e.g. smiling changes all parts of a face. Textures and
> shapes are frequently colocated in real data: an eye of an animal might
> be relevant because of its shape or its color. Dependencies between
> image areas are also significant in real-world datasets, e.g. imagine
> a car crossing a traffic light; the situation would change completely if
> the light were red. Our dataset represents the essence of challenges
> faced in the real world. We will include this in the motivation of the
> dataset in our updated manuscript.
>
> Finally, we want to emphasize that our experimental design is a
> substantial improvement over the state of the art present in the field.
> Most experiments conducted so far do not allow to draw conclusions
> whether users can identify relevant features and discover non-trivial
> biases. They rather evaluate how accessible explanations are. For
> example, users might be able to sort images to a set of concepts
> [(Zhang et al.,
> 2020)](https://arxiv.org/pdf/2006.15417.pdf)
> but this does not reveal if users can reason about how the model uses
> the concepts. In our opinion such results are neither valid nor
> generalizable.
>
> We believe that empirical studies like ours are helpful to guide
> technical advances in interpretable ML. Otherwise, we might risk
> developing techniques not useful to humans. Unfortunately, pre-registered user studies
> that evaluate interpretable deep vision are rare to find, a study design
> based on the bias discovery task is not yet established, nor a required
> data generator with ground-truth feature importances. This year's ICLR
> aims to strengthen such empirical contributions.

---

> > ### Author Response · Authors · 2021-11-29
> > **Please answer to our rebuttal**
> >
> > Dear rVJ7,
> >
> > we posted our initial answers more than two weeks ago and have updated the manuscript last week. However, you did not answer to our rebuttal. We think that we argued well how the experiment protocol could be extended to more complex data domain. Can you please provide a justification for your score? We submitted a valid and faithful manuscript and expect a review process that evaluates our work fairly.
> >
> > Best,
> > the Authors

---

### Author Response · Authors · 2021-11-13
**Overview over Rebuttal**

We first want to thank our reviewers for their time and feedback. Given the high variation in scores, we wanted to get the discussion started as quickly as possible to be able to address all questions and concerns. We already answered each reviewer below and will share the updated manuscript and our code within the next few days.

---

> ### Author Response · Authors · 2021-11-23
> **Update of the Manuscript**
>
> Dear reviewers, we have now updated the manuscript. In particular, we rewrote the introduction to the Two4Two dataset: it now states the motivation of the dataset more explicitly. We also rewrote the concepts’ presentation paragraph and added a paragraph about the generalization to natural data in the conclusion.
>
> We believe that our work has many significant contributions, a view gzkV shares. In our comments from last week, we addressed all concerns raised by the reviewers. In our opinion, much of the criticism is either minor or unfounded. Unfortunately, we have not received a reply from pBBz, 3sBP, and rVJV. While pBBz increased their score, they did not provide a justification. We are looking forward to a more active discussion and are willing to adapt our manuscript based on the reviewers’ suggestions.

---

### Decision · Program_Chairs · 2022-01-20

**Decision:**

Accept (Poster)

**Comment:**

This work presents a novel and clever experiment for interpretable vision.  Reviewers all agreed that it tackles an important and interesting research question via a user study design.  There are some concerns around the generalization and transfer to large-scale real-world settings, as well as dataset construction. With the authors’ responses and discussion, I think the pros seem to outweigh the cons of this work a bit.